# Simplified Modelling of Failure in High Strength Bolts under Combined Tension and Bending

Francesco Plaitano [1,2], Aurel Stratan [1] and Elide Nastri [2,*]

1   Department of Steel Structures and Structural Mechanics, Politehnica University Timisoara, 300224 Timisoara, Romania
2   Department of Civil Engineering, University of Salerno, 84084 Fisciano, Italy
*   Correspondence: enastri@unisa.it

**Abstract:** Bolted connections are widely adopted in steel structures and their behaviour affects to a large extent the global response of the system. High-strength bolts of type HV are commonly employed. Under pure tension, these bolt assemblies usually fail by thread stripping. However, it was observed experimentally that, under combined tension and bending, the failure mode changes to fracture of the shank. The former loading condition commonly occurs in the case of thick extended end plate connections and the latter in the case of flush end plates. In order to analyse the behaviour of the structure, the finite element method (FEM) is usually employed. While there is a wealth of information on FEM modelling of bolts for standard loading conditions (e.g., tension), the authors are unaware of a model able to replicate both tension-only and combined tension and bending conditions. In this paper, a simplified approach to be used in the framework of FEM is proposed to model the behaviour of high-strength HV bolts which can replicate the failure mechanism of bolts under tension only and combined tension and bending. The bolt assembly is modelled with continuum elements, supplemented by a non-linear spring connecting the nut to the bolt shank. The spring captures the stiffness, resistance, and ductility of the bolt-to-nut threaded connection, reproducing the experimentally observed failure mode in the case of pure tension conditions. A simplified damage model is applied to the continuum finite elements used to model the bolt, which replicates shank failure under combined tension and bending as a result of large local stresses and strains occurring under these conditions. The proposed model captures with good accuracy the actual behaviour of high-strength HV bolts under tension only as well as under combined tension and bending.

**Keywords:** FEM; bolted connections; bolt failure; modelling criteria; high strength bolts

## 1. Introduction

High strength bolts are widely used for joining members in steel structures because they offer many benefits, such as high resistance and replaceability of damaged members. Bolted connections are generally made using two types of bolts: HR bolts, designed to obtain ductility predominantly by plastic elongation of the bolt, and HV bolts, designed to obtain ductility predominantly by plastic deformation of the engaged threads [1]. The type of bolts adopted can affect the failure mechanism of the bolted assembly; in fact, there are two main failure mechanisms of those kinds of connections: thread stripping (HV bolts) and shank fracture (HR bolts).

Bolts represent one of the most stressed components of connections, therefore, it is fundamental to evaluate their performance and predict the failure of the connection in advance. The finite element method (FEM) is a convenient tool used to analyse the behaviour of structural assembly subjected to various actions. In fact, FEM permits the showing of the distribution of stresses, plastic deformations, cracks, etc., related to the development of the tests, making it possible to understand the role of each component composing the system. However, to reach an adequate level of accuracy, it is crucial to

model the geometry and the material properties to make it possible to reproduce the actual behaviour that the structural assembly undergoes during the tests.

Standard bolt assemblies are performed under pure tension. Under these conditions, in the case of HV type, the failure usually occurs by thread stripping (Figure 1a). However, when the same bolt assemblies are subjected to combined tension and bending, the failure mode changes to fracture of the bolt shank in the threaded portion (Figure 1b) [2–4]. Mixed shear and axial behaviour are common in bolted connections such as T-stub connections or end plate connections [5–8]. In these cases, it is observed that the main reason for failure is the axial tensile load, even if in large displacements the bending component cannot be neglected. The same problem can be highlighted in the connections of replaceable links, where the bolts undergo large deformations (Figure 1b). In the literature there are some models that can be used to reproduce the stripping of the nut for HV bolts or shank fracture for HR bolts [9–14], but a model capable of fully capturing the behaviour of HV bolts under combined tension and bending is missing [4].

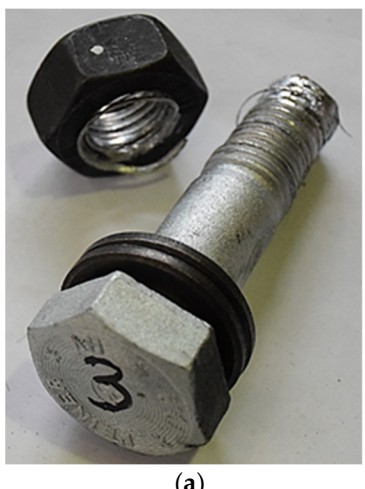
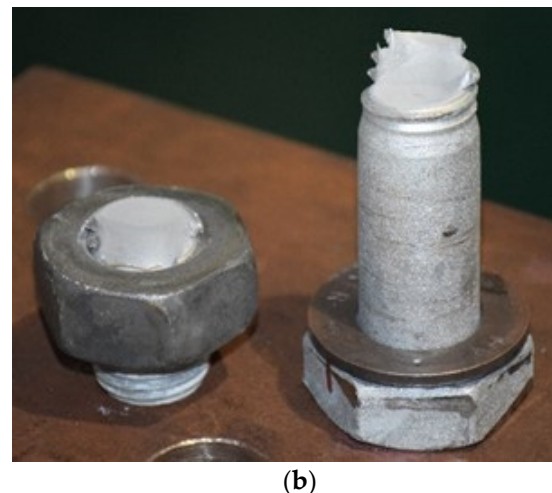

(**a**)  (**b**)

**Figure 1.** Failure of the HV bolt under pure tension (**a**) and under combined tension and bending in a flush end plate connection (**b**) [2,3].

Much research [4–17] was accomplished in recent years to develop models able to replicate, with a good level of accuracy, the damage and failure of bolts subjected to a specific action such as pure tension or shear [9–14], but most of them require the computation of the parameters needed to implement the failure mechanism of the materials and of the joints themselves. Many authors, such as D'Aniello et al. [10,11], proposed a simplified approach to model the failure of European preloadable bolts with the calculation of several damage parameters influenced by many factors, such as the diameter of the bolt, the lengths of the thread and shank, etc., but the computation, calibration and use of these parameters in FEM software like Abaqus [18] may require a substantial increase in time and resources in order to perform the analysis, especially in the case of complex assemblies and cyclic tests [9–17,19–22].

For this reason, a new simplified approach able to model the bolt's behaviour under combined tension and bending was developed without resorting to the fracture mechanics-based models, thus avoiding the increase of the computational time. Several researchers [23–25] investigated the role of damage parameters such as stress triaxiality in FEM simulations, observing that, at present, this kind of parameter is difficult to compute and can significantly affect simulation results, especially under 3-D loading conditions [26].

The proposed approach has the aim of avoiding the computation of damage parameters and simplifying the FEM simulations. The methodology adopted to account for the bolt's bending behaviour starts with the reproduction of the tensile behaviour of M30 class 10.9 HV bolts tested at the University of Timisoara (UPT) laboratory [2,3] in previous re-

search [9–11]. The use of this approach is computationally efficient, achieving the accuracy of more complex methods.

The starting point of this work has been the need to accurately detect the behaviour of bolted flush end plate replaceable links whose experimental tests have been performed at the Politehnica University Timisoara [2,3]. When performing the FEM simulations through Abaqus software, it was highlighted that modelling the bolt behaviour only in terms of axial load was not adequate to correctly catch the ultimate link behaviour and the final fracture point. Therefore, it was necessary to account for the bending behaviour of the bolts.

## 2. Modelling of Bolts in Tension

The model setup started from the results of five bolt tensile tests performed at the Politehnica University Timisoara [3]. The geometry was modelled in Abaqus by considering the bolt, the nut, and the washer as reported in Figure 2. The finite element used is C3D8R (eight-node brick element with reduced integration) with a 5 mm mesh size for all parts. The boundary conditions are applied on two reference points (RP) to replicate the constraints and the actions of the testing machine. In detail, as shown in Figure 2, RP1 is coupled with the inner part of the left washer, while RP2 is coupled with the inner part of the right washer. RP2 is fixed, while a displacement of 10 mm is applied to RP1 along the bolt axial direction. The length of the shank is 61.2 mm with a diameter of 29.8 mm, while the threaded part is 48.6 mm long with a diameter of 25.5 mm.

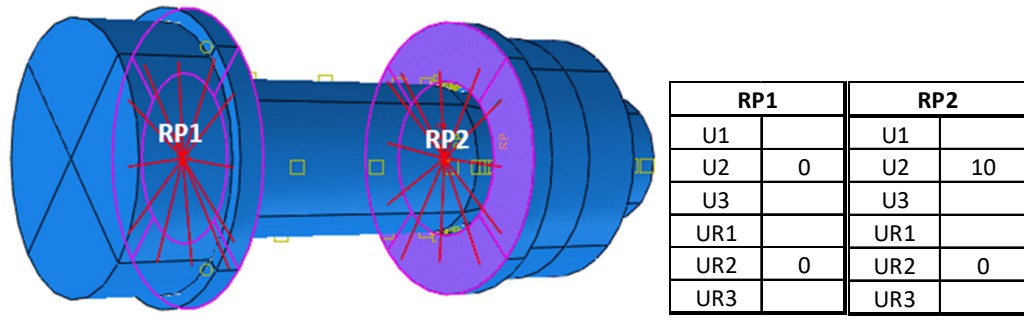

| RP1 | | RP2 | |
|---|---|---|---|
| U1 | | U1 | |
| U2 | 0 | U2 | 10 |
| U3 | | U3 | |
| UR1 | | UR1 | |
| UR2 | 0 | UR2 | 0 |
| UR3 | | UR3 | |

**Figure 2.** Position of the RPs used to apply boundary conditions.

The threaded part has been modelled with a smaller diameter to reproduce the net area section. Regarding the bolt–nut interaction, a non-linear spring element is introduced by means of a wire also considering the plastic damage. Finally, the damage initiation criterion is applied to the plastic material properties. By calibrating the bolt–nut interactions' properties and by properly modifying the plastic stress–strain curve it was possible to account for the bending behaviour.

The nominal material properties are 940 MPa and 1040 MPa for yield stress $\sigma_y$ and $\sigma_u$ ultimate stress, respectively. The mass per unit of volume is 7.85 g/cm$^3$, the Poisson coefficient is 0.3, and the Young's modulus is 210,000 MPa. For the material modelling of the bolt, washer, and nut, the true stress–true strain curve was computed considering a nominal deformation at a rupture of 9%. The resulting plastic stress–strain curve is reported in Figure 3. Moreover, the damage initiation criterion was computed according to [21] with reference to the engineering stress–strain curve in Figure 4. In particular, the equivalent strain at damage initiation has been set as equal to 0.058, while the stress triaxiality parameter, defined as the ratio of the mean stress and von Mises equivalent stress [25], is computed as $\eta = -p/q = 0.33$ [22], where p is the hydrostatic pressure and q is the Mises equivalent stress.

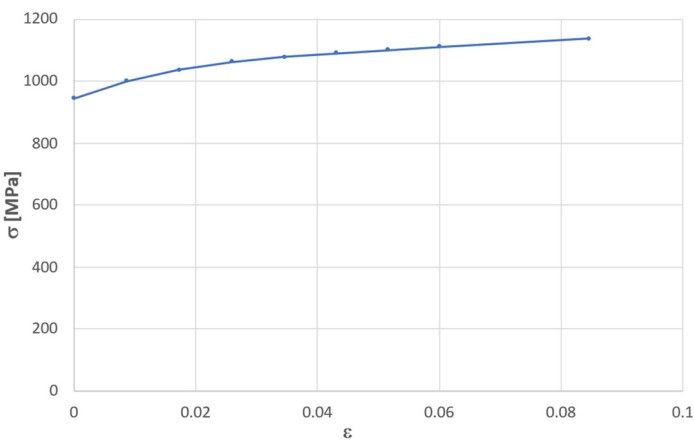

**Figure 3.** Plastic stress–strain curve for steel used in M30 class 10.9 bolts.

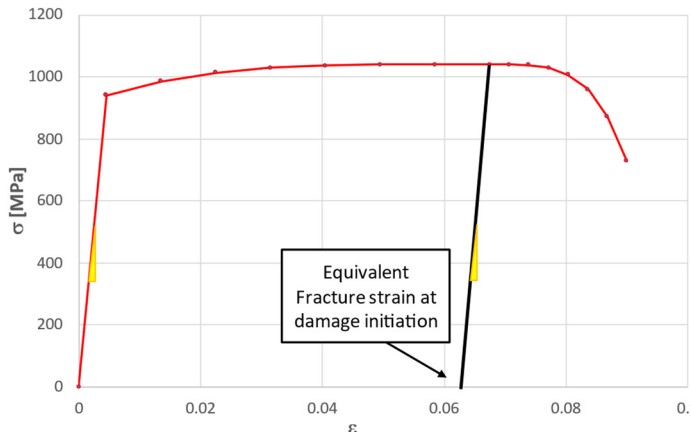

**Figure 4.** Equivalent strain at damage initiation computation.

HV bolts usually fail due to the nut stripping phenomenon, even when they are preloaded. For this reason, it is fundamental to pay attention to the bolt–nut interaction constitutive law to be used in the FEM model. Starting from [9], it was decided to implement an FEM model using a two-pronged approach.

A surface-to-surface contact was established between thread and nut, with tangential frictionless behaviour and normal behaviour with penalty-type hard standard features. Two reference points (RP) were introduced that connect the outer surface of the thread and the inner surface of the nut. They are coupled by using continuum distribution coupling constraint, as shown in Figure 5. The two RPs are, therefore, connected using an axial spring (wire), as depicted in Figure 6.

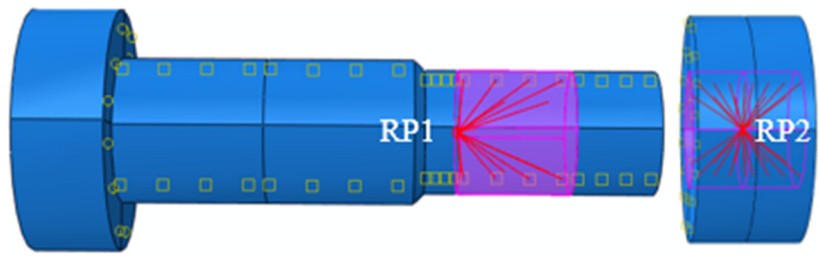

**Figure 5.** Thread and nut reference points (RP).

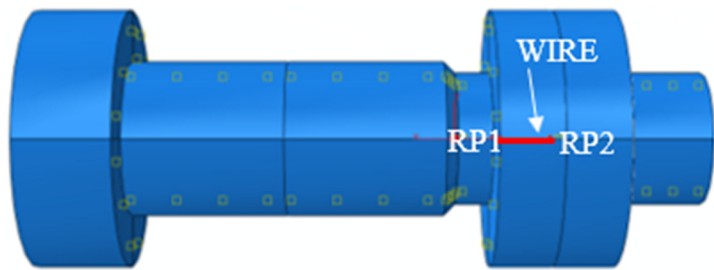

**Figure 6.** Axial spring connecting the reference points RP1 and RP2.

In order to accurately model the constitutive law of the wire connector representing the behaviour of the threaded part, elasticity, plasticity, and damage parameters are computed starting from the experimental tests.

- Elasticity. The initial stiffness is obtained assuming that the behaviour of the bolt–nut assembly is composed of two components acting in series: the bolt shank and the threaded connection between the nut and the shank:

$$K_{th} = \left( \frac{1}{K_b} - \frac{1}{K_{sh}} \right)^{-1} \tag{1}$$

where $K_{th}$ is the initial stiffness of the thread; $K_b$ is the initial stiffness of the bolt–nut assembly; and $K_{sh}$ is the initial stiffness of the shank.

- Plasticity. The plastic deformation was obtained by subtracting the total deformations of the thread from the elastic ones:

$$x_{pl} = x - \frac{F}{K_{th}} \tag{2}$$

where $x$ is the total deformation of the threaded connection, $F$ is the force, and $K_{th}$ is the initial stiffness of the thread.

- Damage. The plastic motion initiation criterion is selected; linear motion damage is specified. To do this, two points must be selected: plastic motion initiation and plastic motion at failure. Both points are kept from the experimental $F - \Delta$ curve: the first one (Point 1 in Figure 7) is the point coinciding with the maximum force achieved, while the second one is the point after which the curve starts to rapidly decrease (Point 2 in Figure 7).

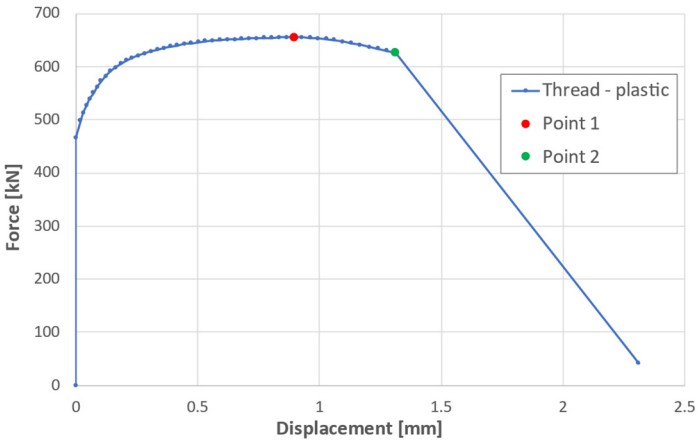

**Figure 7.** Parameters used to define damage to bolt–nut interaction.

The results of the comparison between the experimental and numerical response of an isolated bolt subjected to pure axial tension are reported in Figure 8. It is possible to appreciate the good overlap of the experimental test curve and the FEM simulation. The proposed model makes it possible to achieve a very good grade of accuracy, especially regarding the interaction between bolts and nuts. In fact, replication of the nut stripping phenomenon is obtained, and the stiffness and ultimate force endured by the experimental bolt are adequately reproduced.

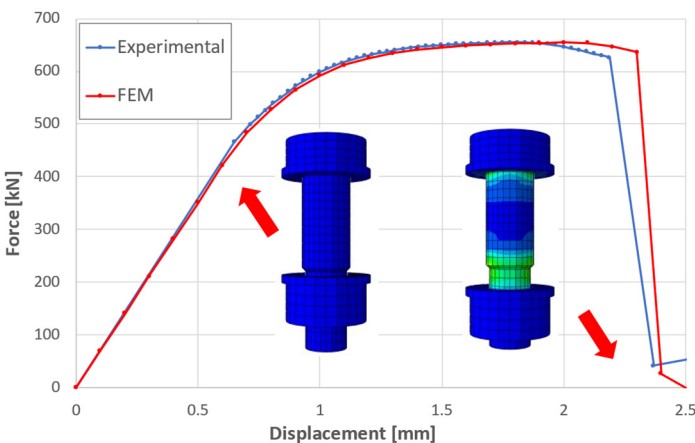

**Figure 8.** Comparison between the experimental and FEM monotonic test on a bolt subjected to nut stripping.

## 3. Modelling of Bolts under Combined Tension and Bending

The novelty point of this work concerns the representation of the behaviour of HV bolts subjected to combined tension and bending. The same procedure adopted for HV bolts subjected to pure tension stress was applied to model the bolt–nut interaction.

In order to achieve correct modelling of bolt behaviour in bending, a comprehensive model for the plastic behaviour of the material is needed. In particular, in order to account for material degradation due to bending moment, a softening branch has been introduced in the plastic material properties as reported in Figure 9. In the literature there are numerous studies regarding the role of damage parameters like fracture strain at damage initiation and stress triaxiality [22–25] for FEM simulations, but the computation of these parameters, in the case of multi-actions, is difficult to attain and the use of them makes the models very sensitive even for very small changes. Starting from this consideration, the calibration process of the plasticity of the material of the bolts started with the aim of avoiding the use of these parameters without giving up the accuracy of the results.

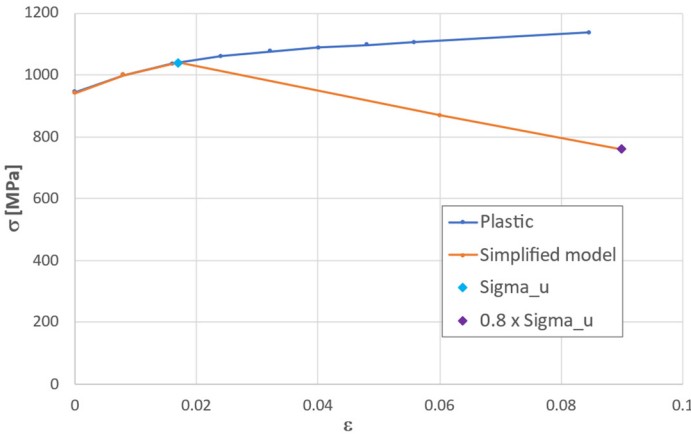

**Figure 9.** Modification of the true stress–true strain curve to account for the material degradation.

A simplified approach to modelling the fracture of the material was developed based on the plastic stress–strain curve previously reported in Figure 3. In this phase, an iterative trial-and-error procedure was used to determine the softening branch of the material to replicate its failure. A modified plastic curve was created which overlaps with the plastic curve previously calculated until the achievement of the maximum nominal tensile stress in the bolt (1040 MPa with a corresponding deformation of 0.017); the second part was calibrated considering the maximum elongation indicated by ISO 898-1 [26] for M30 bolts ($\varepsilon_r$ = 0.09 or 9%). It has been observed that the introduction of a softening branch with a reduction of 20% of the maximum nominal bolt tension (Figure 9) corresponding to an ultimate deformation of 9% is adequate to consider the bolt degradation due to the presence of bending moment. The procedure was validated by using it in FEM models of a flush end plate bolted link and an extended end plate bolted link. From Figure 10 it is possible to observe that the deformed shape of the bolts in the FEM simulations is in good agreement with the experimental tests.

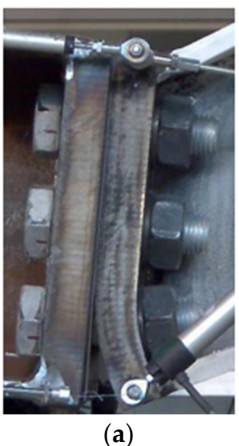
(**a**)

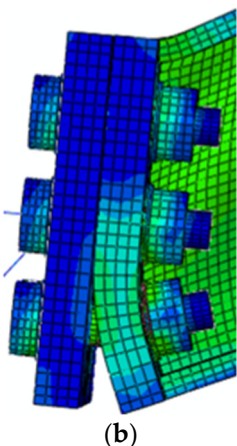
(**b**)

**Figure 10.** Comparison between the deformed shape of the experimental tests (**a**) and the FEM simulations (**b**) where the bending deformation of the bolts is highlighted.

## 4. Discussion

As stated previously, there is no standardized method for testing bolt assemblies subjected to combined tension and bending [27]. For this reason, the effects of the modelling criteria the bolt material and for bolt–nut interaction are observed by using them in the case of a flush end plate bolted link and an extended end plate bolted link. Previous analyses of these assemblies have shown the importance of modelling bolt joints in order to achieve the same behaviour of the numerical model as those shown in the tests [3].

The comparison between experimental and numerical tests of the flush end plate bolted link shows the effectiveness of the modelling criteria adopted to replicate the bolt's behaviour. Figure 11 highlights the failure mechanism of the bolt due to the bending phenomenon, which is well replicated by using the simplified approach proposed in this paper. In fact, in the case of a bolted connection, the role of the bolts is crucial.

Finally, the proposed bolt model was also used in the case of an extended end plate bolted link (Figure 12). Due to the greater stiffness of the extended end plate in comparison with the flush one, bolts are subjected to considerably smaller bending moment in the former case. The predominantly tension failure of the bolt model can be observed. The numerical model was able to reproduce the experimental response well, proving that the developed bolt model can be successfully used for analysis of structural assemblies under tension only or combined tension and bending.

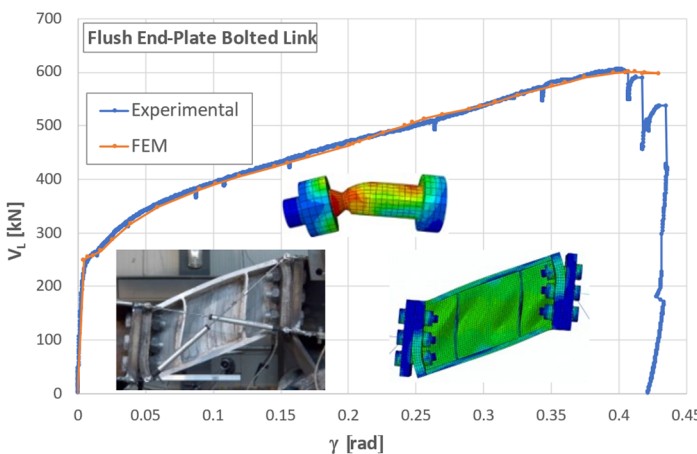

**Figure 11.** Comparison between experimental and FEM monotonic test of a flush end plate bolted link.

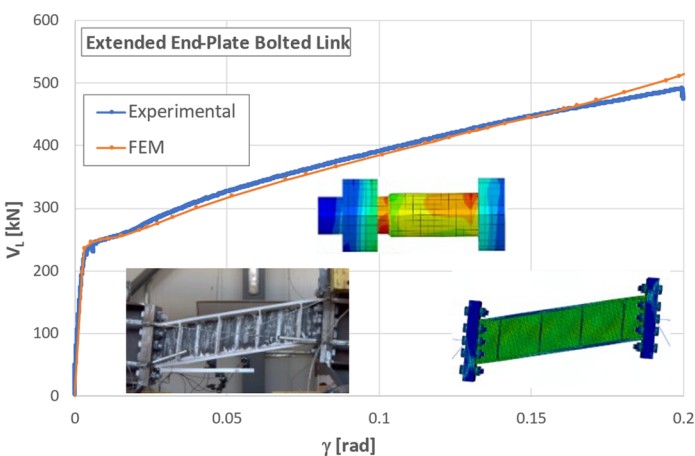

**Figure 12.** Comparison between experimental and FEM monotonic test of an extended end plate bolted link.

## 5. Conclusions

The main objective of this paper was the development of a simplified approach of modelling failure of grade 10.9 HV bolts under combined tension and bending. Experimental tests revealed two different failure modes: thread stripping in the case of pure tension and shank fracture in the case of combined tension and bending. The simplified numerical model consists of the combination of a three-dimensional model representing the bolt where an axial spring is used for the modelling of the thread stripping. A variable shank diameter and simplified bolt material damage capture the strain localization and subsequent fracture of the shank under tension and bending of the bolt. As testified by the simulated tests, the proposed model can adequately capture the combined tension and bending of bolt assemblies. Moreover, it allows for reduction of the computational time, as it does not resort to complex degradation modelling for the material properties.

**Author Contributions:** Conceptualization, A.S. and F.P.; methodology, A.S., F.P. and E.N.; software, F.P.; validation, F.P., A.S. and E.N.; formal analysis, F.P.; investigation, F.P.; resources, A.S.; data curation, F.P.; writing—original draft preparation, F.P.; writing—review and editing, E.N. and A.S.; visualization, F.P.; supervision, A.S. and E.N. All authors have read and agreed to the published version of the manuscript.

**Funding:** This work was supported by a grant of the Romanian Ministry of Education and Research, CCCDI–UEFISCDI, project number PN-III-P2-2.1-PED-2019-5427, with-in PNCDI III. The support is gratefully acknowledged.

**Institutional Review Board Statement:** Not applicable.

**Informed Consent Statement:** Not applicable.

**Data Availability Statement:** Not applicable.

**Conflicts of Interest:** The authors declare no conflict of interest. The funders had no role in the design of the study; in the collection, analyses, or interpretation of data; in the writing of the manuscript; or in the decision to publish the results.

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
