# Peer review of "Simplified Modelling of Failure in High Strength Bolts under Combined Tension and Bending"

_jcs, doi:10.3390/jcs6100302_

Round 1

Reviewer 1 Report

The authors presented the work by simplifying the damage experienced in bolts in FE simulation that enables capturing their performance. The research concept is nice. but the manuscript is poorly written. The reviewer just highlighted a little portion in the attachment, where those statements are awkward with many grammar issues, throughout the entire manuscript. The reviewer suggested the authors should totally revise the entire manuscript, starting with the title, abstract, and each section. 

Also, please add the findings in the abstract (where you should present why you do this research, what are your methods, and what are results. The conclusion section did not show the contribution of this study.

Round 2

Reviewer 1 Report

The authors provided most response through the manuscript and the reviewer suggest revising the abstract with more findings before accepting it.

Reviewer 2 Report

The authors have revised their paper, however, the refs number in the main text needs to be updated, they did not apper in sequence.
